# Cross-Cultural Adaptation and Validation of the Spanish HLS-COVID-Q22 Questionnaire for Measuring Health Literacy on COVID-19 in Peru

**DOI:** 10.3390/healthcare13151903

**Published:** 2025-08-05

**Authors:** Manuel Caipa-Ramos, Katarzyna Werner-Masters, Silvia Quispe-Prieto, Alberto Paucar-Cáceres, Regina Nina-Chipana

**Affiliations:** 1Escuela Profesional de Ingeniería Comercial, Facultad de Ciencias Jurídicas y Empresariales, Universidad Nacional Jorge Basadre Grohmann, Tacna 23000, Peru; mcaipar@unjbg.edu.pe; 2Faculty of Business and Law, Manchester Metropolitan University, Lyceum Place, Manchester M15 6BY, UK; a.paucar@mmu.ac.uk; 3Escuela Profesional de Enfermería, Facultad de Ciencias de la Salud, Universidad Nacional Jorge Basadre Grohmann, Tacna 23000, Peru; squispep@unjbg.edu.pe (S.Q.-P.); rninac@unjbg.edu.pe (R.N.-C.)

**Keywords:** cross-cultural adaptation, COVID-19, health literacy, Peru, Spanish, pandemic, psychometric properties

## Abstract

**Background/Objectives:** The social importance of health literacy (HL) is widely understood, and its measurement is the subject of various studies. Due to the recent pandemic, several instruments for measuring HL about COVID-19 have been proposed in different countries, including the HLS-COVID-Q22 questionnaire. The diversity of cultures and languages necessitates the cross-cultural adaptation of this instrument. Thus, the present study translates, adapts, and validates the psychometric properties of the HLS-COVID-Q22 questionnaire to provide its cross-cultural adaptation from English to Spanish (Peru). **Methods:** As part of ensuring that the final questionnaire accommodates the cultural nuances and idiosyncrasies of the target language, the following activities were carried out: (a) a survey of 40 respondents; and (b) a focus group with 10 participants, followed by expert approval. In addition, the validity and reliability of the health instrument have been ascertained through a further pilot test administered to 490 people in the city of Tacna in southern Peru. **Results:** The resulting questionnaire helps measure HL in Peru, aiding better-informed decision-making for individual health choices. **Conclusions**: The presence of such a tool is advantageous in case of similar global health emergencies, when the questionnaire can be made readily available to support a promotion of strategies towards better self-care. Moreover, it encourages other Latin American stakeholders to adjust the instrument to their own cultural, language, and socio-economic contexts, thus invigorating the regional and global expansion of the HL study network.

## 1. Introduction

Whilst the main purpose of health literacy (HL) is to facilitate making judgments and decisions regarding healthcare, disease prevention, and health promotion to achieve a better quality of life [1,2], the concept itself has evolved and shifted in the last couple of decades from being a determinant of health [3] to an integrative concept entailing the knowledge, motivation, and competencies to access, understand, appreciate, and apply health information. Evidently, this definition makes the association between HL, on one side, and population health problems, inadequate medical services, obstacles to access to healthcare, as well as a greater probability of death, on the other side. This highlights the relevance of measuring HL and emphasises the importance of evaluating the hindrances to understanding health issues, with the goal of identifying initiatives that could increase the level of HL. This is important since improved health literacy leads to higher and sustainable levels of well-being [2,3,4], thus benefitting individuals as well as health systems and governments [5].

Yet, evidence suggests that levels of HL are low worldwide. For instance, Europe has poor health literacy rates ranging between 29% and 62%, with Austria and Bulgaria characterised by the lowest rates [6]. Likewise, in research carried out in Mexico, 8 out of 10 people with diabetes had low levels of HL [7]. Similarly, in a Brazilian analysis of the relationship between HL and quality of life in adults diagnosed with arterial hypertension, over 82% of participants were reported to have unsatisfactory HL [8]. Whilst the studies focusing on HL in Latin America and the Caribbean are scarce and condition-specific, they persistently highlight largely unsatisfactory results [9,10,11]. Peru is amongst the Latin American countries with a prevalence of low health literacy levels [10].

With the appearance of COVID-19, a transmissible infectious disease caused by the SARS-CoV-2 virus, being health-literate and staying well-informed proved crucial yet challenging for the prevention of and reduction in the transmission of the pandemic [12]. The overload of information—together with a spike in the volume of false information about the pandemic through the internet and social media networks—increased the individual complexity associated with making appropriate self-care and health choices [13,14]. This so called infodemia [15] created the need to measure HL so that reliable information could be distinguished from misinformation, thus lessening the vulnerability to COVID-19.

Indeed, Okan et al. [16] highlight the critical importance of health competence during the COVID-19 pandemic due to the uncontrolled proliferation of information. Their study investigates the health skills and capacity of adults who use the internet, emphasising the importance of improving skills to make informed decisions and reduce apprehension linked to the pandemic. In particular, it proposes the HLS-COVID-Q22 questionnaire as a feasible and reliable tool to assess coronavirus-related health literacy in adults in Germany [16]. The proposed tool is a modified version of the HL questionnaire created earlier as part of the European Union Literacy Survey Project (HLS-EU-Q), focused on measuring HL amongst its members [1,17]. The HLS-COVID-Q22 questionnaire includes 22 items organised to test individual competencies to access (six items), comprehension (six items), evaluation (five items), and the application of information (five items).

To the best of our knowledge, the only study that provides an instrument for testing health literacy in the context of COVID-19 in Spanish is Falcón et al. [18], who mainly focus on assessing the psychometric properties of the 9-item COVID-19 Health Literacy Questionnaire (CHL-Q) in Spain. However, the authors do not outline the exact procedure describing the linguistic adaptation of the CHL-Q instrument from English to Spanish. We believe this stage of the adaptation process is important as it affects the validation of the instrument, leading to more accurate results in terms of the self-reported HL. In addition, the questionnaire they validate is different to the one proposed in the present study as it contains only 9 instead of 22 items to assess the four types of competencies related to HL. Finally, the psychometric analysis conducted by Falcón et al. [18] is representative of the Spanish (as in Spain only) general population and as such, might not extend to other Spanish-speaking populations globally. This creates the need to carry out an adequate cross-cultural adaptation [19] of the relevant HL instrument to other Spanish-speaking environments.

The adaptation of measurement tools related to HL, in different cultural and linguistic contexts, contributes to evaluating the development of HL and the relevance of its use in those different contexts and regions. Specifically, in reference to the linguistic translation, it has been recognised that, to guarantee the representation of the original language in the destination, a simple translation is not enough. Thus, a back translation must be added that allows for the differences to be captured and overcome to become idiomatically representative [20]. Such a rigorous adaptation process provides confidence to the user, increasing the positive impact of informed decision-making on the part of public health authorities [21,22].

Furthermore, the cross-cultural adaptation process—including psychometric testing of instruments measuring HL—is a task that contributes to understanding the health needs of the target population, leading to a better provision of high-quality medical care and the promotion of health [23,24]. As argued by Meyer et al. [25], comprehensive validation studies are needed to ensure the feasibility of a novel measurement tool. Recognising that these aspects are even more important in the context of developing countries—where healthcare provision differs from that in developed economies [14]—this study aims to translate, culturally adapt, and psychometrically validate the HLS-CVID-Q22 questionnaire into Spanish for the Peruvian context.

Using the application of the cross-cultural adaptation process outlined by Sousa and Rojjanasrirat [26], we adapt the HL questionnaire from English to Spanish (Peru) by accommodating cultural nuances and linguistic idiosyncrasies of the target language. Subsequently, we analyse psychometric properties of the adapted instrument using a pilot test of 490 participants in Tacna, Peru, to ensure the validity and reliability of the final version of the questionnaire.

The choice of Peru as the target country is not arbitrary. This country has been recognised as having the highest COVID-19 death rate in the world, with over 6400 deaths per million of citizens [27]. Thus, the adapted and validated instrument helps measure HL in Peru, where such an instrument has not been available previously to facilitate making appropriate health choices. This further contributes to an improved healthcare system that supports individuals in making informed decisions regarding their health and healthcare specialists in providing their services. We believe that the adaptation of the instrument to the Peruvian context can encourage other Latin American stakeholders to adjust the instrument to their own cultural, language, and socioeconomic contexts, thus invigorating the regional and global expansion of the HL study network. In addition to facilitating better-informed health choices in Peru, the presence of such a tool is advantageous in case of similar global health emergencies, when the questionnaire can be made readily available to support the promotion of strategies towards better self-care by the healthcare sector and relevant authorities.

## 2. Materials and Methods

This section describes the methodology pertaining to the process of cross-cultural adaptation of the health literacy questionnaire to Spanish (Peru), with the emphasis on linguistic translation and psychometric testing. Ethical considerations are also addressed.

### 2.1. Study Design and Settings

This study is part of a larger project, the ALSAVI project, which aims to measure HL in the Viñani area in the Tacna region of southern Peru. The results were shared with the relevant authorities, who are willing to support the next step of the project, namely the application of the adapted health questionnaire.

The English version of the HLS-COVID-Q22 instrument [1] was the one that underwent the cross-cultural adaptation process described by Sousa and Rojjanasrirat [26] (see also Beaton et al. [28] and other authors [20,29,30,31,32]). The process combines a conceptual–qualitative treatment to solve possible linguistic (local and regional) disagreements and the use of quantitative techniques with current technology to assess the psychometric properties of the instrument [33]. Specifically, the analysis focuses on content validity linked to the translated version of the instrument as well as construct validity and reliability tests to assess the psychometric properties of the questionnaire.

The process consists of seven steps (see Figure 1), which are split into two phases. The operational phase involves the first five steps, designed to linguistically adapt the instrument to the target language (in our case Spanish), and was completed in December 2022. The last two steps focus on psychometric analysis of the linguistically adapted instrument to determine its validity and reliability. The last step was completed in early 2023.

It is worth mentioning that the timing of the data collection was not random. Data was collected following the third wave of the pandemic and after the end of the general lockdown (27 October 2022). At the time, the epidemiological situation in Peru had slightly improved, with a total of over 216,000 deaths and less than 30 deaths daily compared with over 200 daily casualties in February 2022 [34]. The use of masks became optional in open spaces and in ventilated enclosed spaces as of 1 October 2022. This enabled the research team to engage with participants as part of the focus group supporting the linguistic translation of the HL instrument.

In Figure 1: SL stands for source language; TL for target language; PI–TL for preliminary initial target language; B–TL for back translation language; and P–FTL for pre-final target language.

In the following sections, we describe the two phases of the cross-cultural adaptation process of the HLS-COVID-Q22 instrument and the underlying study approach in more detail.

#### 2.1.1. Linguistic Translation

The original HLS-COVID-Q22 instrument consists of 22 items divided into four themes focused on measuring people’s competencies to access (6 items), understand (6 items), evaluate (5 items), and apply (5 items) information about COVID-19 to make informed health choices. Two translations of the instrument from English into Spanish (Peru) were produced in early December 2022 by two professional translators with Spanish as their mother tongue and relevant subject knowledge.

In a subsequent review, an independent (third) translator and a panel—consisting of six experts specialising in linguistics—compared the two translated versions, leading to the first synthesis with the first version of the questionnaire in Spanish.

Then, two blind back translations from Spanish to English were carried out mid-December 2022 by two native English translators. These back-translated versions of the instrument were then compared as part of the second synthesis, where the panel of experts discussed possible discrepancies between them.

Finally, a pilot test of the pre-final version of the instrument in Spanish with a Peruvian sample was conducted. The role of this pilot test was not to quantitatively validate the instrument but to add the element of transculturality into the questionnaire through the analysis of phrasing, which embodies the specific idiomatic nuances of the language. For this purpose, two techniques were applied: (i) a quantitative technique involving a survey of 40 respondents to determine the clarity of the questions; and (ii) a qualitative technique entailing a focus group with 10 participants (different to the survey’s respondents) to detect categories that improve the writing of the items.

For the survey, 40 people were selected from the Tacna region, including participants from the Viñani sector. The majority of them (80%) were women, aged 18 and over. All have expressed their consent to participate in this study and were clearly informed of the study’s purpose, its protocol, and the voluntary nature of their participation. Once the survey responses were processed, they were analysed to feed into the discussion in the focus group.

The focus group consisted of 10 participants, different from those completing the survey. They were invited to physically participate in the focus group meeting, which took place in a community setting for ease of reach. The participants were provided with information about this study, informed consent was obtained, and the objectives as well as the techniques to be used in the focus group were explained.

First, the questionnaire was administered individually to the focus group’s participants, allowing them to become familiar with the instrument. A group session was then held, in which they were asked to comment on their understanding and interpretation of each statement, as well as to suggest improvements. Specifically, questions were asked, such as “*Was the question understood?*”, “*What does it refer to?*”, “*Do you think the item should be maintained or modified?*”, and “*What do you propose?*”. Participants actively collaborated and, although most understood the items, they noted that some wording and language could be improved.

Although the session was not recorded, the researchers kept detailed written records as participant observers. The activity lasted ninety minutes. Finally, the notes were reviewed, transcribed, and the information coded to identify categories, patterns, and conclusions that reflect the study’s objective.

Based on the above, a thematic analysis of the interventions was conducted to identify patterns of comprehension, semantic difficulties, and suggestions for improvement.

Once the thematic analysis was completed with the suggested changes to the instrument, the panel of experts became involved. The panel was made up of seven health professionals, all with substantial professional experience and expertise relevant to this study. They reviewed and validated the wording of the items, considering the findings which emerged from the survey and the focus group, and prior to the psychometric testing.

The process helped produce the linguistically adapted questionnaire, which accounts for aspects of the Spanish language that include cultural nuances and idiosyncrasies of the destination.

#### 2.1.2. Psychometric Analysis

With respect to the psychometric analysis, a cross-sectional, observational study was conducted on a stratified sample of 490 participants in Tacna, Southern Peru, in early 2023. All participants were 18 or older and Peruvian. The participants reported their age, gender, and the area of residence.

The analysis followed the quantitative approach and instrumental design of Ato et al. [35] to ensure the validity and reliability of the final instrument in Spanish. In terms of criterion validity, specifically concurrent validity, we sought to correlate the adapted Health Literacy in Environment COVID-19 (HLS-COVID-Q18) instrument with the SAHLSA-50 (Short Assessment of Health Literacy for Spanish Adults) instrument developed by Lee et al. [36], which measures general health literacy.

As a preliminary step to the factor analysis, a descriptive analysis of items was performed to evaluate the consistency of their distribution (centralisation, dispersion, and shape) and their contribution to the overall consistency measurement of the instrument. In the first step, Exploratory Factor Analysis (EFA) was applied in compliance with the initial Kaisser–Meyer–Olkin (KMO) tests, Barlett’s sphericity, and using polychoric correlations and methods appropriate for ordinal data. The 2-, 3-, and 4-factor models were analysed, respectively. Subsequently, Confirmatory Factor Analysis (CFA) was applied to determine the fit of the translated instrument, define its possible respecification, and interpret the results [37,38,39,40]. The application of both EFA and CFA procedures allowed for the development of an adapted version of the instrument based on statistical rigour, added to the element of conceptual equivalence [41] resulting from the preceding linguistic translation procedure.

The analysis was facilitated by the open-source tool Jeffreys’s Amazing Statistics Program (JASP), a well-known software based on C++ programming code, R, and Javascript, which allows for working with ordinal data, as in this study.

Given the lack of multivariate normality identified through the Mardia test and a sample size greater than 200, the use of the Diagonally Weighted Least Squares (DWLS) method was ratified as pertinent [42,43]. The magnitudes of the fit indices given by the JASP software were corroborated by code in open-source software R/R Studio using the lavaan library [44] that facilitated creating code for the polychoric matrix, from which the fit indices were generated (We have used the following software versions: JASP 0.16.4, released 3rd October 2022; C++ version C++23, released in 2023; R, version R 4.3.1 released 16 June 2023; Javascript version ES2023 publised June 2023; RStudio version 2023.03, released March 2023; Lavaan, version 0.6-15 released April 2023).

For the fit indices, current robust cut-off values were used [45,46]. The first group of (absolute) fit indices met the value requirements of Chi square/degrees of freedom (χ^2^/df) < 3; Standardised Root Mean Residual (SRMR) close to 0; Root Mean Square Error of Approximation (RMSEA) < 0.08; Goodness-of Fit-Index (GFI) ≥ 0.95. The second group of (incremental) fit indices satisfied the following: Normed Fit Index (NFI) ≥ 0.95; Tucker–Lewis Index (TLI) ≥ 0.95; and Comparative Fit Index (CFI) ≥ 0.95 [47].

As the indices with our data did not show appropriate adjustment with respect to the model (structure) of the original linguistically translated instrument (RMSEA values and Chi-square ratio χ^2^/df outside the accepted range), the respecification procedure continued in search for a structure that meets the indices within the required ranges [48]. Two criteria were followed. The first criterion was operational, considering high modification indices, and facilitated eliminating the items with the lowest factor loading in each pair of items. The second criterion was the usefulness of the meaning captured by an item to respondents and helped remove items that were not meaningful. The product of such considerations defined a respecified model that generated an instrument that maintains the original four factors, with a reduction from 22 to 18 items.

To ensure the reliability of the final instrument, internal consistency and factorial invariance analyses were conducted. The former was implemented to provide basic psychometric evidence that reflects the replicability of the instrument. Given the four-option Likert scale employed in the translated instrument, this was performed using an ordinal alpha index of greater relevance [49,50,51] as well as the omega index due to its applicability in the context of polychoric matrices [52,53]. Both indices consider the specific factor loading that each item of the questionnaire contributes to its factor.

To verify the internal consistency of items corresponding to each factor, the Fornell–Larcker [54] composite reliability criterion (CR) was applied. To verify the discriminant validity, the HTMT (Hetero-Trait–Mono-Trait) method was used as the most appropriate alternative. The required values are 0.7 to 0.9 for CC [55] and 0.90 and 0.85 for HTMT [55,56].

The invariance analysis was also carried out to determine if the adaptation of the instrument preserves the assumption that groups of men and women internalise the same meaning for the questionnaire items of each dimension. Hence, lavaan syntax [44] was run in JASP, confirming that the instrument measures the same constructs in both groups. Then, the invariance by sex was verified in the sample using the mean difference test.

Given the outline of the cross-cultural adaptation process, some ethical considerations when involving pilot testing with human participants are also discussed.

### 2.2. Ethical Considerations

It is important to acknowledge that this comprehensive multi-step process must be rigorously followed to produce the desired outcome and as such, entails relevant ethical considerations. Hence, an approval from the Research Ethics Committee of the Jorge Basadre Grohmann National University was sought and subsequently granted to conduct this study. To ensure ethical conduct of this study, permission was sought and received from the original authors of the questionnaire to use the instrument for cross-cultural adaptation purposes as part of the mentioned ALSAVI project, which aims to adapt the instrument to Spanish to measure health literacy in the Viñani area in the Tacna region of southern Peru. Finally, all participants involved in this study were appropriately briefed about the nature of their participation and signed their informed consent accordingly. The data was collected anonymously by administering paper versions of the survey to the participants.

## 3. Results

The purpose of this section is two-fold. First, the results of the operational phase that focused on the linguistic translation are presented in Section 3.1. These are followed by showcasing the findings of the psychometric testing in Section 3.2, where the reliability and validity of the adapted instrument are assessed.

### 3.1. Linguistic Adaptation

Completing the first four steps of the cross-cultural adaptation process [26] allowed the original health questionnaire in English (see Figure 2) to be translated into Spanish. An analysis confirming the final phrasing of the items, the clarity of wording, and the semantics derived from it followed as a form of content validation. For this purpose, a pilot survey of 40 respondents was conducted, which was followed by a focus group with 10 participants. The feedback was considered, and additional suggestions were made by the panel of experts, who evaluated the instrument.

The survey enabled the respondents to provide feedback regarding the clarity of the semantics of each question/item. Accordingly, the 40 participants ranked the clarity of each of the questionnaire’s items (“clear” or “unclear”) and made additional observations. In total, 10.2% of participants attributed the characteristic “unclear” to the items and indicated the reason behind it. Once the survey responses had been processed, the researchers considered the use of some terms that were highlighted, such as “probability”, “evaluate”, “coronavirus”, among others. The survey’s results fed into the discussion in the focus group.

In the application of the focus group technique, the suggestions raised through the survey’s responses were considered by 10 participants. This process helped identify the expressions supposedly confusing the semantics of the questions and the corresponding item. Specifically, a thematic analysis of the interventions was conducted to identify patterns of comprehension, semantic difficulties, and suggestions for improvement. Two main categories emerged from this analysis. The first category, the clarity of language, focused on (i) wording that could be improved, and (ii) the use of more common synonyms instead of unfamiliar terms. The second category, the adaptation of terminology, featured the need to adapt certain expressions to better reflect the sociocultural context of Tacna. Hence, the terms “coronavirus” and “instructions” were replaced with the more widespread expressions used in the target population, namely, “COVID-19” and “indications”, respectively. The term “radio” was added to item 03 as it is a commonly used means of communication in the Peruvian family environment. Some words have been removed to reduce the length of expressions where their omission does not affect the meaning but improves readability. Finally, the group of experts evaluating the final structure of the instrument made additional suggestions. For example, it was decided that the problematic verb “Judge”, translated to “Evaluate”, would be kept (items 13–17). In this way, the formal, linguistically translated Spanish version of the instrument was obtained with semantics that include the cultural nuances and idiosyncrasies of the target language (Figure 3). This final version was tested using the Aiken test, giving it a value of 92.05%—as an expression of a high content validity—confirming that the produced instrument accurately reflects the concept being measured, prior to the psychometric testing.

### 3.2. Psychometric Properties

To complete the cross-cultural adaptation process, the psychometric analysis of the linguistically adapted HLS-COVID-Q22 instrument in Spanish was conducted in the final step. The purpose of this analysis was to determine the validity and reliability of the questionnaire. To achieve this, the questionnaire was completed by a sample of respondents. We characterise this sample next.

#### 3.2.1. Sample Characteristics

The questionnaire was distributed, and the information regarding its purpose and voluntary participation was clearly outlined. Subsequently, 490 respondents completed the survey, with 239 men (48.8%) and 251 women (51.25%). All participants were over 18 years of age, with a mean of 41.12 and standard deviation of 5.98, where 50.8% of respondents were below 40, and 49.2% were age 40 or above.

The instrument was distributed to five different areas with different population sizes. A total of 30.2% of participants from the most populated area and 8.2% from the least populated area completed the survey, with each of the three remaining areas accounting for approximately 20% of completed questionnaires.

#### 3.2.2. Fit Indices and the Validity of the Internal Structure

As an expression of criterion validity, which was represented in the concurrent validity score, a moderate Pearson correlation of 42.2% was reported. The descriptive analysis of items showed values of skewness and kurtosis that are in the acceptable range of −1.5 to 1.5, whilst the corrected item–total index complies with values > 2. This shows that the items, in general, do not show atypical states in their position, dispersion, or shape measures, thus, contributing to the consistency measurement of the instrument.

In application of the EFA procedure, the KMO estimate was verified and gave a value of 0.9. Barlett’s sphericity, which rejects the identity matrix, was used, and estimators were selected for ordinal data. Models with two, three, and four factors were analysed, respectively. The cases with two and three factors reported an accumulated variance lower than 0.5. When analysing the case of four factors coinciding with the hypothesised model, the cumulative variance of 0.53 was reported. The EFA allowed us to identify items Q1, Q2, Q12, and Q20 with anomalous loadings close to or higher than 1 (a sign of possible multicollinearity problems), and items with insufficient loadings such as Q5 and Q14.

Subsequently, CFA was applied to analyse fit indices. The analysis of fit indices was carried out with the JASP statistical tool using the WLSMV estimator, appropriate for matrices that do not have multivariate normality as in our case. In addition, we followed the rigorous statistical principle of not abusing derived respecifications and of being guided only by the removal of items due to residual covariances. This was achieved through searching for acceptable values of fit indices under standard cut-off points and reviewing the phrasing that guides the decision on the item remaining. Conceptually, a removal of an item occurred in the presence of another item with equivalent meaning but semantically more general and/or with less time-dependent phrasing.

The residual covariance of pairs of items suggested by the software was considered and those with higher modification index values were analysed, as follows. The analysis of the residual covariance of items 01 (“Search for information on the internet about COVID-19”) and 02 (“Search for information in newspapers, magazines, radio, or television about what to do to prevent contagion by COVID-19”) revealed the need for further analysis of the phrasing of both items. Ultimately, the elimination of item 02 was determined, given that item 01 is semantically more general and maintains greater stability over time with respect to the concept of the pandemic after the global experience.

Item 08 (“Understand the recommendations of the authorities on how to protect yourself from contracting COVID-19”) was withdrawn following the analysis of the covariance of residuals of this item, and item 07 (“Understand the instructions of your doctor, pharmacist, or nurse on how to protect yourself from contracting COVID-19”), with the latter having a higher factor loading.

Similarly, item 12 (“Understand the information given in newspapers, magazines, radio, or television about how you could be infected with COVID-19 again”) was withdrawn in favour of item 10 (“Understand the information given in the media about how you could be infected with COVID-19 again”), which shows less time-dependent phrasing. Additionally, the word “again” was removed from item 10 to account for the possibility of some not being infected before.

Finally, the analysis of residual covariance of items 19 (“Follow the instructions of your doctor or pharmacist on how to manage a case of COVID-19”) and 20 (“Use the information given by your doctor to decide how to control a COVID-19 infection”) led to the removal of item 20, with the former item being representative of both. This and previous adjustments have been summarised in Table 1, which shows that, in line with the exclusion of some items oriented by the EFA process, a reduction from 22 to 18 items was implemented that did not affect the 4-factor structure of the instrument.

Figure 4 shows the final 18-item instrument with its initial numbering. The values that accompany the central arrows are the factor loadings of acceptable magnitude. The lower values are the residual errors. The values of the upper arrows correspond to the correlation between the factors.

Table 2 compares the fit indices capturing the absolute and incremental adjustments of the originally adapted model (HLS-COVID_Q22—Model 1) and the respecified model with 18 items, outlined in Figure 4, Model 2. The matching values were obtained with JASP and R.

It is evident that indices X^2^/gl, RMSEA, and SMRM do not provide good fit values with respect to the required cut-off points [45] for the originally cross-culturally adapted questionnaire with 22 items (see the first row of Table 2). However, this changes when the number of items is reduced to 18 (respecified model) so that good values of fit indices are obtained for the sample (Peruvian) data.

This process, supplemented with the evidence of conceptual equivalence derived from the linguistic translation, provides a final step of the validation of the cross-culturally adapted HL instrument, with 18 instead of 22 items. This 4-factor instrument is shown in Figure 5.

#### 3.2.3. Reliability

To verify the internal consistency of the respecified model, the ordinal alpha was calculated using Excel and confirmed using a code in the R/R Studio language (psych library), to compare the index calculated from the numerical matrix with that of a polychronic one. This verifies whether the values of each factor are undervalued when the ordinality of data is not considered. The residual covariance of pairs of items suggested by the software was considered, and those with higher modification index values were analysed, as follows. The omega index was also calculated, where the factor loadings provided by the items were used as input.

The test for the reliability of the respecified model showed good results, as indicated in Table 3. Compared to the values for the factors between 0.66 and 0.74 of Cronbach’s alpha (α)—which considers continuous rather than ordinal data and equal factor loadings—the factors with ordinal alpha obtain high values between 0.76 and 0.81. The corresponding values of the factors are 0.81 for “access”, 0.76 for “understand”, 0.77 for “appraise”, and 0.76 for “apply”. Similarly, these factors take values of 0.76, 0.68, 0.7, and 0.67, respectively, when the omega index is calculated. Both indices (Cronbach’s alpha and omega) consider the factor loading of each item with respect to its respective dimension.

The application of the composite reliability criterion (CR) reported the following indices: access = 0.82, understand = 0.76, appraise = 0.77, and apply = 0.76; these results indicate that the groups of items consistently measure the same construct to which they belong (Table 3). The test of discriminant validity using HTMT shows all the indices below the 0.9 threshold. More conservatively, all the indices are below the 0.85 threshold, except for access–understand (0.857) and appraise–apply (0.888).

In order to verify that the instrument measures the latent variable in the same way between men and women, factorial invariance analysis was carried out. This analysis has confirmed that the instrument expresses the same meaning in both groups. Specifically, Table 4 shows that the null hypothesis (the intersections of the items are equal in the two groups) cannot be rejected (*p* > 0.05) for Model 3 (scalar invariance). This is performed using Model 1 (configurational invariance) as a benchmark. This result shows that the structure of the model is useful in verifying the equality of average scores between the groups, meaning that the wording of the items generates the same meaning amongst men and women. This outcome is statistically acceptable, even though the adjustment of Model 2 (metric invariance) does not allow us to firmly accept that the factor loadings are equal in the two groups (*p* < 0.05). An opposite result has been obtained for the factorial invariance analysis by age groups.

To verify the factorial invariance resulting from the test for groups by gender, verification was carried out in the sample with the respecified model using Student’s t test, with the null hypothesis of no difference in the average scores of both groups. The hypothesis was not rejected (t = 1.65 and *p*-value = 0.1), suggesting that there was no difference in the average scores of men and women. As mentioned above, an equivalent test for age groups supported the idea that differences by age groups are present (t = 4.56 and *p*-value < 0.05).

## 4. Discussion

The aim of producing the adapted version of the questionnaire was to facilitate the process of measuring HL in Peru during a health crisis, thus, aiding better-informed decision-making for patients’ health choices. Ultimately, the intention behind the generated instrument is to use it in Spanish-speaking environments to increase the level of health literacy of the target population. The low levels of HL worldwide and in Latin America in particular, as well as the uncertainty following the lifting of confinement measures and the appearance of new strains of the coronavirus, call for a greater promotion of HL. A positive aspect of these outcomes that will allow measures to be taken is that, in recent years, there has been a vast and rapid growth of research in HL [1]. This shows that HL does not only constitute an important objective of healthcare systems and services in Latin America [11] but also in some European countries [6,16].

The present paper contributes to the increasing literature focusing on COVID-19 HL by adapting the HLS-COVID-Q22 questionnaire to Spanish, through the application of the back-translation procedure and psychometric testing as part of a rigorous process of cross-cultural adaptation. In the following, the findings of these two stages are discussed and compared to those in the existing literature. The limitations of the present study are also considered.

### 4.1. Discussion of the Linguistic Adaptation of the HL Instrument

Linguistic translation of an instrument is a crucial step in the cross-cultural adaptation process. Its role is to preserve the intention behind the original instrument available in the source language when translated to a target language, thus overcoming the challenges arising from different cultural and linguistic patterns. If misconducted, it can compromise both the validity and reliability of the translated instrument [57].

Many authors agree that the most recommended procedure to propose a cross-cultural equivalence of an instrument translated from one language to another is back translation [6]. Such a process allows for overcoming the differences in language and customs, making the instrument idiomatically representative [29,32].

Indeed, this research employed the linguistic translation process—part of the broader cross-adaptation procedure proposed by Sousa & Rojjanasrirat [26], Beaton et al. [28], and other authors—to adapt the HLS-COVID-Q22 questionnaire from English to Spanish. The rigorous linguistic translation process conducted in this study consisted of five steps, with the final step embracing a pilot study for content analysis. This step involved (i) a survey of 40 respondents, and (ii) an additional focus group with 10 participants. In this way, the final questionnaire was designed to accommodate the cultural nuances and idiosyncrasies of the target language.

We found that the survey proved to be a useful tool in clarifying the final phrasing of the questionnaire’s items and verifying the clarity of wording and semantics. Indeed, 4 out of 40 participants indicated some items as “unclear” and explained their reasoning behind this judgement. These items involving terms, such as “probability”, “evaluate”, and “coronavirus”, among others, were discussed as part of the thematic analysis conducted with the help of the focus group.

The application of the instrument to a focus group with the target population helped ensure content validity through practical adequacy [58], which made it possible to specify terminology. Indeed, the experience of finding differences in the connotation, meaning, and understanding of terminology corresponds to the fact that there is disparity between beliefs, behavioural norms, and social habits of different populations and cultures. Using language as a proxy for cultural origin, Deopa and Fortunado [59] show that exposure to COVID-19 is impacted by culture. This idea, with language featuring the cultural distinctiveness, is further supported by Li [60]. In our case, the cultural discrepancies arose from using the original instrument designed for the German population and adapting it to the Peruvian population. Through the rigorous procedures carried out, cross-cultural equivalence was secured by responding to the criteria of content, meaning, understanding, and conceptualisation [19], captured through pilot testing and the involvement of experts’ judgement.

It is worth mentioning that Falcón et al. [18] also assess the validity of a Spanish version of the COVID-19 Health Literacy Questionnaire. Whilst they explore the psychometric properties of the instrument in the Spanish population, their study pays little attention to the linguistic translation. Similarly, Dubova et al. [7], who apply the HL instrument to evaluate the association between the health literacy of people with Type 2 diabetes and their health outcomes, do not pay explicit attention to the translation of the instrument. In their study, the Spanish version of the European Health Questionnaire was applied in the context of diabetic patients without undergoing prior cross-cultural adaptation. However, it is known that the linguistic translation step is crucial in ascertaining the validity and reliability of the instrument, enabling it to reach a more diverse and broader sample of respondents [61] through the cross-cultural adaptation process. Thus, by paying explicit attention to the linguistic translation of the cross-cultural adaptation process, the present study emphasises the importance of this stage for subsequent psychometric testing and the overall accuracy of the COVID-19 health measures.

### 4.2. Discussion of the Psychometric Testing of the HL Instrument

Whilst linguistic translation has not received much attention in the literature studying the cross-cultural adaptation process, some research lacks details of the psychometric analysis of the properties of the instrument used [62]. Here, this has been addressed with Structural Equation Modelling (SEM) that admits the review of covariance structures between latent variables or factors and measurable or reactive variables [63]. In addition, factor analysis and the availability of software tools, in a continuous production of new versions, make it possible to manage the validity of instruments through psychometrics with statistical processes [64].

In the present paper, the psychometric measurement process facilitated the assessment of adjustment indices. As they did not meet the appropriate values, residual covariance analysis was employed to modify indices of high scores applied to pairs of items: 1 and 2; 7 and 8; 10 and 12; as well as 19 and 20. This led to a respecification of the instrument in that items 2, 8, 12, and 20 were removed. This is an important result that is consistent with the findings obtained by Okan et al. [16], who, using SPSS AMOS 22 software, pointed out that the fit indices of the unmodified (original) model “suggest insufficient fit”. Like in our case, the residuals of some pairs of items were correlated. However, according to our analysis, the fit indices were favourable in line with the more robust criteria of Hooper et al. [47], with required values greater than 0.95 for GFI, CFI, and TLI. This positive result of our adaptation process can be explained by using software that allows us to work with polychoric matrices (by having an ordinal Likert scale of four for responses). SPSS versions used in the study by Okan et al. [16] could not work effectively with such matrices.

A point of methodological usefulness that further strengthens our contribution, and is noted by other authors [65], is that the final removal of items does not strictly obey the process suggested by the statistical technique for the possible elimination of items. Rather, it is combined with the discretion of the project team in that the semantic usefulness of said items requires their conservation. For instance, item retention in the present analysis occurred in the case of items 16 and 17, suggested for removal by the software. However, the team kept the items since the phrasing of item 16 expressed the possibilities of becoming infected, that is, a situation when the risk factors that facilitate contagion are known. Item 17 complements item 16 by evaluating whether one has been infected/knowledge of symptoms or diagnostic tests/whom to reach out for a diagnosis. Hence, given that the two items semantically address two different points in the natural history of the disease, the suggestion produced by the software was rejected.

Variants of our study are shown in other countries, such as in Norway with the HLS-Q12 instrument, with an acceptable fit to the unidimensional Rasch model, achieving acceptable goodness-of-fit indices using CFA [66]. In a Brazilian study evaluating the psychometric properties of an oral health literacy assessment by Bado et al. [67], a unidimensional model was used with an explained variance of 71.23%, adequate levels of factor loading, communalities and item discrimination, as well as stability and replicability of the instrument to other populations. In general, however, these models show problems with the adjustment of the four-factor structure assuming a unidimensional model [16].

Regarding reliability, the majority of studies focusing on testing psychometric properties of HL instruments use Cronbach’s alpha to assess their reliability (e.g., [67,68]). The present study used the ordinal alpha test instead of Cronbach’s alpha since the latter tends to underestimate reliability for ordinal response scales with fewer than five alternatives, like in our case. The application of the former test showed internal consistency with high values of ordinal alpha between 0.76 and 0.81 for the factors. This indicates that the items of the instrument consistently measure the variable.

For the composite reliability, indices were found that support the items’ belonging to each factor. The results are aligned with the discriminant validity, which indicates that the factors are related but conceptually differentiated. However, the access–understand and appraise–apply factors may suggest a high latent correlation since their indices (0.857 and 0.888) exceed the conservative threshold of 0.85.

In practice, the usefulness of supporting software that allows somewhat more personalised control of some procedures must be recognised. In our case, we added the support of the open-access software R/R Studio (lavaan library) to the use of JASP software to explicitly generate the code to create the polychoric matrix, to then include the appropriate “WLSMV” estimator. The latter takes into account the ordinal nature of our matrix, which is not numerical as automatically assumed by other software. Compared to packaged software, customising this concurrence (type of matrix/type of estimator) leaves no doubt regarding the appropriateness of the fit indices in the present model.

Towards the end of the adaptation process, a suggestion resulted from monitoring twenty-five quality control points in the cross-cultural adaptation process by Hambleton and Zenisky [58]. Specifically, a point referring to possible changes in grammar and wording was highlighted for item 10, with the need to eliminate the word “again” (in Spanish *nuevamente*, see Figure 3) following the elimination of item 12. In this way, item 10 captured the meaning intended by both items.

Likewise, the APA reporting guide for Quantitative Research Design (JARS-Quan) was taken into consideration, referring to the inclusion of SEM elements such as the primary model, global fit indices, factors, and multivariate normality analysis.

Thus, the present study led to a Spanish version of the HLS-COVID-Q22 questionnaire that can be easily understood and readily used. As Spanish is one of the most spoken languages in the world and the main language in Latin America, the present study offers a ready-made tool for policy makers focused on improving the health literacy of patients in those countries.

### 4.3. Limitations

Whilst extensive care has been taken to ensure that the produced instrument provides a reliable and valid measure of health literacy in the context of COVID-19, certain shortcomings must be mentioned. These are considered next.

One of the limitations of this study is the possibility of sample bias [69]. Whilst the goal of developing the COVID-19 HL questionnaire in Spanish is to benefit the Peruvian population, we recognise that using a sample from Tacna might limit the generalisability of our results due to its specific cultural and linguistic characteristics. Tacna is a region with a population representative of two of Peru’s three geographic zones. Whilst methodologically, the sample of 490 is sufficient for the analysis, geographically it may not be representative of the entire Peruvian population, indicating a potential bias in the sample selection.

Another bias that must be acknowledged in the context of the present study is the self-reporting bias [70]. Okan et al. explicitly state that the HL questionnaire is a self-reported measure, and it is not intended to evaluate health literacy capabilities based on individual performance. Thus, by choosing an instrument that relies on self-reported data, our questionnaire becomes more susceptible to this bias. However, as with all self-reported HL measures, it is difficult to balance between the cost-effectiveness of such screening methods and their reliability [71]. Given the widespread prevalence of the self-reported measures, compared with other alternatives (e.g., performance-based tests) in the context of COVID-19, we can only conclude that these instruments are considered effective in assessing people’s HL and, respectively, in designing appropriate interventions. However, we acknowledge that special attention must be paid to this bias, particularly when social and educational vulnerabilities might affect the accuracy of such health literacy measures.

Further limitations apply to the methods employed in the psychometric testing phase. As already reported, in terms of criterion validity, specifically concurrent validity, we sought to correlate the adapted Health Literacy in Environment COVID-19 (HLS-COVID-Q18) instrument with the SAHLSA-50 instrument [36] to ascertain the concurrent validity of our instrument. Although the result showed moderate correlation, suggesting that both instruments share an acceptable amount of variance consistent with expected concurrent validity, due to time constraints and the longitudinal aspect of this study, it was not possible to perform a predictive analysis with respect to criterion validity.

The time aspect also impeded the test–retest analysis, which resulted in an inability to better assess the reliability characteristics of the adapted instrument. Specifically, given the sequence of this study (linguistic adaptation followed by sample application and construct validation) and the time constraints, it was not possible to carry out the test–retest procedure, which would have complemented the reliability and temporal stability analyses. However, we intend to incorporate this test in future research, and are aiming at applying the instrument to examine HL in the Viñani area of the Tacna region.

Given these limitations, further research on cross-cultural adaptations of instruments—particularly in the context of assessing psychometric properties of the instrument in socio-educationally vulnerable contexts—should, as a continuous dynamic process, allow for the sharing of measurement tools in the spirit of achieving democratisation or universal access to a better quality of life worldwide.

## 5. Conclusions

Measuring COVID-19 health literacy is crucial for both Latin American and global populations, as there is still uncertainty about the virus and its future evolution. Although in 2023, the WHO deemed COVID-19 as no longer a public health emergency of international concern, it insists that the disease continues to pose a global threat. More so, as we speak, new coronavirus variants are emerging, highlighting the possibility of novel viruses triggering future pandemics. These concerns increase the need to have reliable and validated instruments to quickly evaluate and improve the population’s understanding of health issues. Systematic measurement of HL levels in local and regional contexts is a starting point for developing effective health information access and communication strategies.

This is precisely where the contribution of the present study sits. This study represents the first COVID-19 HL instrument with cross-cultural adaptation of the HLS-COVID-Q22 questionnaire (origin), into Spanish, for the Peruvian and possibly other Latin American contexts. Using a sample from the Tacna region in southern Peru, this study develops a COVID-19 HL instrument that facilitates individual choices regarding self-care and helps the relevant authorities understand how people handle news related to COVID-19 to better design effective interventions addressing the aftermath of the pandemic. This is particularly relevant for developing countries, which, like Peru, are characterised by the presence of significant barriers to healthcare access and improved health outcomes at all levels.

Although the selection of study participants specifically from the Tacna area is associated with certain constraints, as mentioned earlier, the main purpose of employing the outcome for this research is for the questionnaire to be integrated into public health policies as part of the ALSAVI project in the Viñani area in the Tacna region of southern Peru. This will then allow for the implementation of more focused and efficient programs that promote health literacy in this vulnerable community in a sustainable manner.

Indeed, as part of the ALSAVI project and in co-operation with the local leaders and health authorities, the developed questionnaire will be applied in the Viñani community to measure health literacy in the province. The results will be used to design more tailored interventions that are culturally and linguistically appropriate and specific to community needs, thus reducing health inequalities in this region of Tacna. One of the main goals is to identify the community members with limited health literacy, to better support them in improving their health outcomes, thus reducing healthcare costs. Given the links with other provinces of Tacna and beyond, it is foreseen that the method created for measuring health literacy in this study could be replicated in other regions of Peru to improve general health and reduce disparities in the access to and understanding of health information.

Therefore, the COVID-19 health literacy instrument developed in this study is a useful technical tool for the formulation of evidence-based regional public policies in Peru, aimed at improving the quality of the relationship between health services and the community. Its application should further improve adherence to medical treatments and promote healthy lifestyles.

Beyond the direct impact of informing the design of health policies in the Viñani province of Tacna in Peru, this line of research is also enabling us to connect with organisations in other countries with whom we have established agreements and participated in conferences on HL topics to further disseminate the findings of this research. Ultimately, the goal is to improve the quality of life of individuals through better-informed decision-making and disease prevention.

## Figures and Tables

**Figure 1 healthcare-13-01903-f001:**
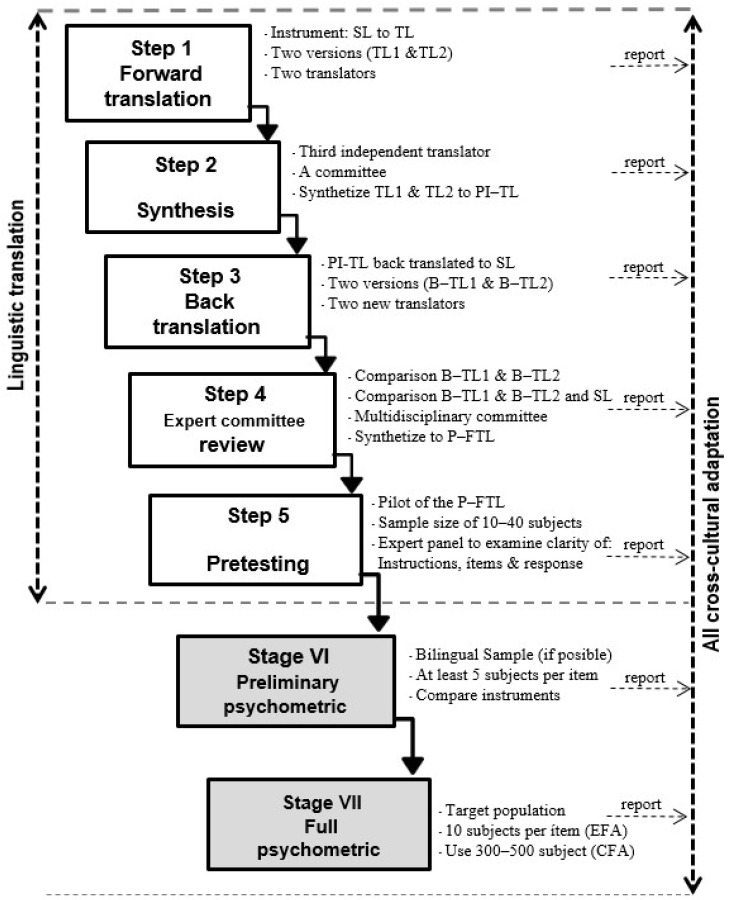
Cross-cultural adaptation process (adapted from [26]).

**Figure 2 healthcare-13-01903-f002:**
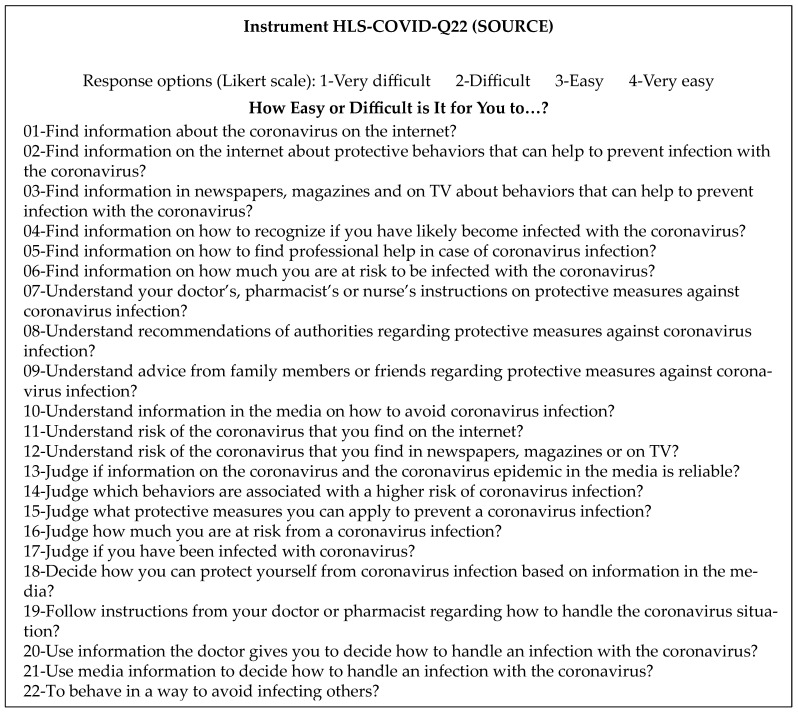
Original HLS-COVID-Q22 [1].

**Figure 3 healthcare-13-01903-f003:**
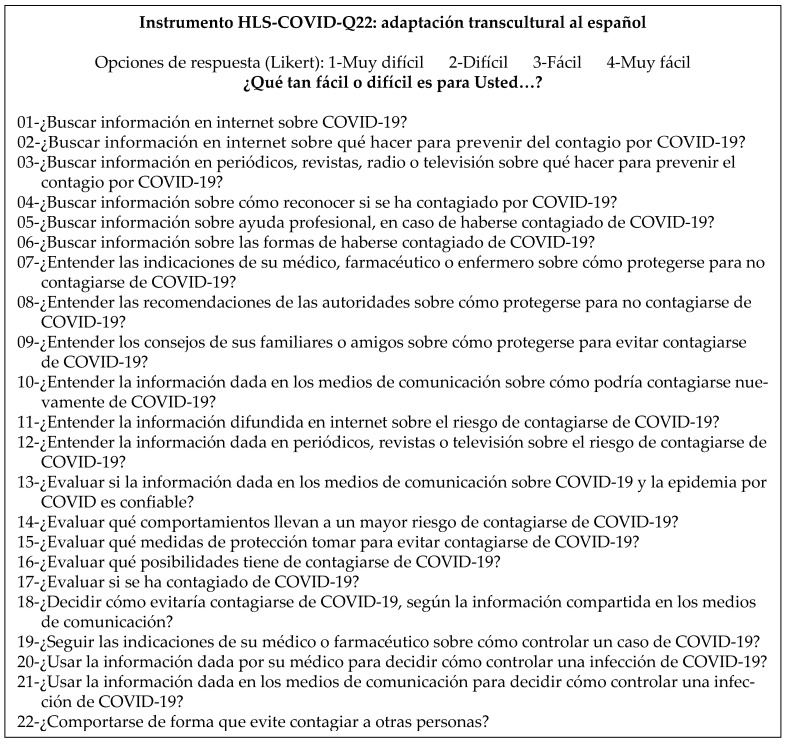
HLS-COVID-Q22 instrument, linguistically adapted to Spanish.

**Figure 4 healthcare-13-01903-f004:**
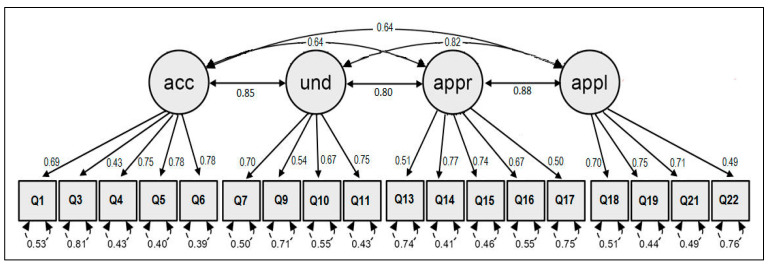
Factor structure respecified to 18 items, with items 2, 8, 12, and 20 removed; acc = access; und = understand; appr = appraise; appl = apply.

**Figure 5 healthcare-13-01903-f005:**
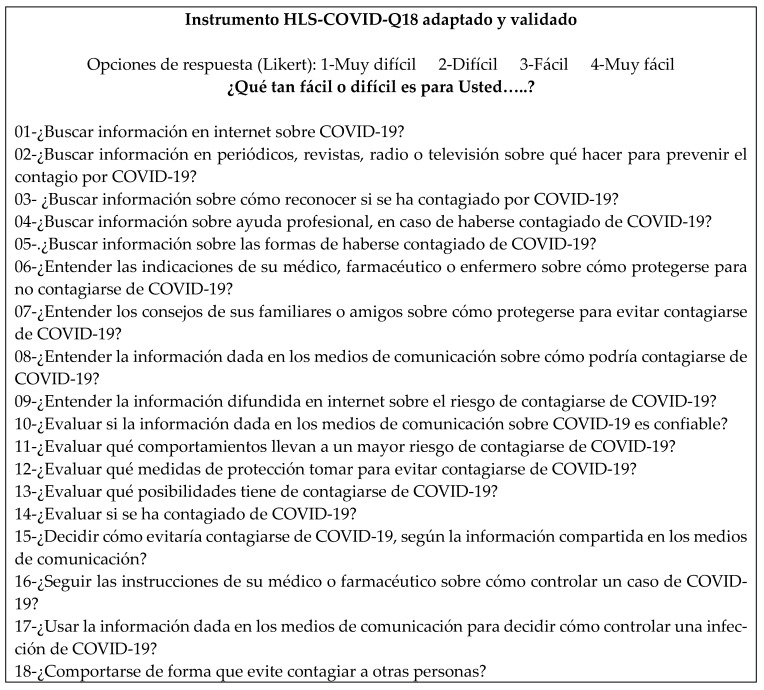
Model 2 respecified (HLS-COVID-Q18).

**Table 1 healthcare-13-01903-t001:** Initial and final instrument structure, by adaptation.

Original: 04 Factors and 22 Items	Respecified: 04 Factors and 18 Items
Factor 1: 6 items (1,2,3,4,5,6)	Factor 1: 5 items (1,3,4,5,6)
Factor 2: 6 items (7,8,9,10,11,12)	Factor 2: 4 items (7,9,10,11)
Factor 3: 5 items (13,14,15,16,17)	Factor 3: 4 items (13,14,15,16,17)
Factor 4: 5 items (18,19,20,21,22)	Factor 4: 5 items (18,19,21,22)

**Table 2 healthcare-13-01903-t002:** Fit indices for the original and respecified models.

Adjustment		Absolute Fit				Incremental Fit		
Measures		X^2^/gl	RMSEA	GFI	SRMR	NFI	CFI	TLI
Model 1	Original	4.461	**0.084** (0.079–0.090)	0.977	0.078	0.964	0.972	0.968
Model 2	Respecified	2.969	0.063 (0.056–0.071)	0.984	0.063	0.969	0.979	0.976

**Table 3 healthcare-13-01903-t003:** Summary of factors, loadings, alpha and omega, composite reliability.

Factor	Indicator	λ	Resid-e	α	Ord-α	Ω	CR
Access				0.74	0.81	0.76	0.82
	Q1	0.69	0.53				
	Q3	0.43	0.82				
	Q4	0.75	0.43				
	Q5	0.78	0.40				
	Q6	0.78	0.39				
Understand				0.66	0.76	0.68	0.76
	Q7	0.71	0.50				
	Q9	0.54	0.71				
	Q10	0.67	0.55				
	Q11	0.75	0.43				
Appraise				0.68	0.77	0.70	0.77
	Q13	0.51	0.74				
	Q14	0.77	0.41				
	Q15	0.74	0.46				
	Q16	0.67	0.55				
	Q17	0.50	0.75				
Apply				0.68	0.76	0.67	0.76
	Q18	0.7	0.51				
	Q19	0.75	0.44				
	Q21	0.71	0.49				
	Q22	0.50	0.76				

λ = factor loading; α = Cronbach’s alpha; Ord-α = ordinal alpha; Ω = omega; CR = composite reliability.

**Table 4 healthcare-13-01903-t004:** Fit table for factorial invariance, by gender.

				Baseline Test			Difference Test		
	AIC	BIC	*n*	χ^2^	df	*p*	Δχ^2^	Δdf	*p*
Model 1	15,011.942	15,515.270	490	562.137	258	<0.001			
Model 2	15,011.267	15,455.874	490	589.462	272	<0.001	27.325	14	0.017
Model 3	14,998.792	15,367.900	490	612.987	290	<0.001	23.525	18	0.171

## Data Availability

The information generated in the linguistic adaptation process of the original HLS-COVID-Q22 instrument comes from (1) the report with results of translations from English to Spanish, translation synthesis, back translations from Spanish to English, and a pre-final version to the target language; (2) observations of a sample of the population (focus group) on the semantic clarity of the 22 questions (which were later analysed and clarified by the research committee); and (3) the result of an expert judgment evaluation as a final contribution to the adaptation process. The information can be requested from the corresponding author.

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
