# Peer review of "Cross-Cultural Adaptation and Validation of the Spanish HLS-COVID-Q22 Questionnaire for Measuring Health Literacy on COVID-19 in Peru"

_healthcare, 2025, doi:10.3390/healthcare13151903_

Round 1

Reviewer 1 Report

Comments and Suggestions for Authors

Thank you for the opportunity to review this manuscript. It has been a valuable and relevant academic exercise.

The article addresses a highly topical and practically relevant theme, offering a significant contribution to the cross-cultural adaptation of a well-established international instrument for assessing health literacy (HL) in the context of COVID-19. The overall design of the study is solid, and the authors demonstrate a good understanding of the methodology for linguistic and cultural adaptation.

However, the statistical validation of the instrument requires further attention in order to meet current scientific standards in the field of psychometrics. In the absence of essential analyses to support the robustness of the instrument’s validity, the conclusions presented in the study may appear premature or insufficiently grounded in empirical evidence. These aspects are discussed in detail in the following sections of the review.

I consider the paper to have strong publication potential, but it requires substantial improvements to meet the level of scientific rigor expected in an international journal with a medical and methodological profile.

Introduction

  • I noticed an informational overload in paragraphs 5–8, where too many ideas are combined (definitions of HL, its importance, international studies, and the need for cultural adaptation). A clearer structure or division of content—particularly regarding the state of HL in Europe and Latin America—would make the text easier to follow.
  • As currently constructed, the introduction does not clearly state the objective of the article, especially for readers unfamiliar with the specific field. While the direction of the research is implied, the explicit research aim is missing. A clear sentence such as “The aim of this study is to translate, culturally adapt, and psychometrically validate the HLS-COVID-Q22 questionnaire into Spanish for the Peruvian context” should be added.

Materials and Methods

  • It is unclear who exactly constituted the pilot sample. The authors mention 40 respondents and 10 participants in a focus group, but do not specify demographic criteria or education levels. It would have been important to explain whether the focus group was recorded, how thematic analysis was conducted, etc.
  • The section on statistical validation is generally well structured and demonstrates commendable technical rigor. The authors correctly apply confirmatory factor analysis (CFA) using the DWLS estimator suitable for ordinal data and rely on modern software tools (JASP and R/lavaan). Furthermore, internal consistency was assessed through both ordinal alpha and omega coefficients, which provides a solid psychometric foundation.

However, several significant methodological omissions undermine the robustness of the instrument’s overall validation:

  1. The absence of convergent and discriminant validity analyses, such as AVE (Average Variance Extracted) or the Fornell–Larcker criterion, is notable. In confirmatory factor models, such analyses are essential to demonstrate that the instrument’s dimensions are distinct and conceptually well defined.
  2. Criterion validity—whether concurrent or predictive—is not tested. Given that HL is often linked to external variables such as education level or health behaviors, correlating scores with such variables would have greatly strengthened the instrument’s practical value.
  3. No detailed item-level analysis was conducted (e.g., item-total correlations, item discrimination indices). Such analyses are standard in any psychometric validation process and help clarify each item’s contribution to its respective factor.
  4. It is unfortunate that the authors did not carry out a test–retest procedure to assess temporal stability. This is essential if the instrument is to be used for monitoring HL in clinical or educational settings over time.
  5. The lack of an exploratory factor analysis (EFA) prior to CFA is also a concern. While the theoretical structure of the instrument is known, in the context of a cross-cultural adaptation to a region with distinct linguistic and social characteristics, conducting an initial EFA would have been advisable to empirically validate the latent structure.

I strongly recommend that the above analyses be incorporated into a revised version of the manuscript to ensure the psychometric robustness and practical utility of the HLS-COVID-Q18 scale.

Regarding the ethics subchapter, it is not clear whether data collection was anonymous or coded. Additionally, the data collection method is not described (was it paper-based? online? in what context was the questionnaire administered?). These details should be added for full transparency.

Results

The results section is clearly structured in two main parts: the linguistic adaptation of the HLS-COVID-Q22 instrument and the psychometric validation of the Spanish version adapted for the Peruvian population. Both components reflect an evident concern for methodological rigor, but several important limitations require correction or further elaboration to qualify the instrument as scientifically validated.

  • Concerning the linguistic adaptation, the authors follow the classic forward/backward translation model, with expert review and pilot testing (n=40) plus a focus group (n=10). It is commendable that they provide examples of vocabulary changes adapted to the Peruvian socio-cultural context. However, this section lacks important methodological details: the composition of the expert committee is not specified, demographic data of pilot participants are missing, and there is no systematic documentation of the qualitative analysis of the focus group (e.g., identified categories, illustrative examples, or how specific items were reworded).
  • The statistical validation is conducted via CFA using DWLS, appropriate for ordinal data. It is a strength of the paper that modern software tools (JASP and R/lavaan) are used, and that model assumptions (e.g., Mardia’s test, N=490) are explicitly mentioned. The initial 22-item model showed inadequate fit (RMSEA=0.084, χ²/df > 4), but the revised 18-item model achieved excellent fit indices (RMSEA=0.063, CFI=0.979, TLI=0.976). The adjustments are explained both statistically (residual covariances) and semantically (retaining conceptually relevant items).
  • Nevertheless, despite these promising elements, the results omit several essential analyses considered standard for psychometric validation, as outlined in the previous section (e.g., convergent, criterion, and stability validity). The absence of these tests significantly limits the strength of the validation evidence presented.

Thus, while the authors demonstrate an advanced understanding of psychometric validation and correctly apply CFA and reliability testing, the lack of convergent, criterion-related, and temporal stability analyses limits the scientific strength of the instrument in its current form.

Discussion

  • The discussion is overly focused on restating the methodology, at the expense of interpreting the findings. Several paragraphs (e.g., 1–3 and 5–6) repeat information from the methods section without expanding the discussion into a broader or comparative context. At times, the discussion reads as a duplicate of the previous sections, which weakens its analytical function.
  • There is a lack of comparative analysis with similar studies in the region. Although European and some Latin American studies (Brazil, Mexico) are cited, there is no systematic comparison between HL levels reported in this study and those from other validations of the HLS-COVID-Q22 or similar instruments in Spanish-speaking countries.
  • Key methodological limitations are not addressed, such as the absence of critical validation tests (convergent validity, test–retest, EFA), which directly affect the strength of the study’s conclusions. The possibility of sampling bias (490 respondents from a single region) is not discussed, nor are the limitations of self-reported HL data in vulnerable socio-educational contexts.
  • The practical implications are overly idealized and vague. The authors suggest that the instrument could be used in public policy, doctor–patient communication, or reducing health inequalities—but they do not provide concrete scenarios or examples of integration into healthcare or education systems.

Although the authors manage to integrate their findings into a coherent and regionally grounded narrative, the discussion section is dominated by repetition and reaffirmation of the cultural adaptation value, rather than by critical analysis of limitations or in-depth reflection on psychometric validity. Clarifications regarding potential limitations, international comparisons, and concrete examples of practical application are necessary. Most importantly, the discussion should be updated to reflect any new statistical analyses conducted in response to earlier recommendations. As it stands, the discussion lacks depth and requires significant revision to support the study’s conclusions credibly.

Author Response

Dear Reviewer 1,

Thank you for your comments and please find attached our replies.

Best wishes

Reviewer 2 Report

Comments and Suggestions for Authors

This manuscript presents a valuable and well-structured study focused on the cross-cultural adaptation and psychometric validation of the HLS-COVID-Q22 health literacy instrument in the Peruvian context. The topic is timely and relevant, particularly in light of the global emphasis on health literacy following the COVID-19 pandemic. The study is methodologically grounded, combining both qualitative (focus groups, expert review) and quantitative (CFA, reliability testing) components, which strengthen the overall rigor. However some issues must be improved:

Item reduction rationale: While respecification improved model fit, the theoretical justification for item removal (beyond statistical criteria) could be elaborated to ensure the conceptual fidelity of the final instrument.

Reliability scope: The internal consistency measures are sound, but the study would benefit from reporting confidence intervals and potentially including test-retest reliability to assess temporal stability.

Generalizability: Since the sample is limited to Tacna, more discussion is needed on how the instrument may (or may not) generalize to other regions of Peru with distinct cultural or linguistic contexts.

Practical implications: The manuscript could be strengthened by offering clearer pathways for how this instrument will be used in public health decision-making or intervention planning in Peru

-How culturally and linguistically appropriate is the adapted Spanish version of the HLS-COVID-Q22 for the Viñani population?

-How representative and engaged were the focus group participants involved in the linguistic adaptation?

-To what extent does the post-pandemic timing of the data collection influence participants’ health literacy perceptions and responses?

-How will the results of this adapted instrument inform future public health interventions or policy decisions in the Tacna region? It should be discussed.

There is a probable problem. The direct linguistic translation alone was insufficient to preserve the original questionnaire's psychometric structure in this new cultural context, necessitating adaptation beyond language to ensure validity. This may reflect cultural differences in interpreting certain items or a need for further qualitative refinement. How this impact has been considered?

Comments on the Quality of English Language

This manuscript presents a valuable and well-structured study focused on the cross-cultural adaptation and psychometric validation of the HLS-COVID-Q22 health literacy instrument in the Peruvian context. The topic is timely and relevant, particularly in light of the global emphasis on health literacy following the COVID-19 pandemic. The study is methodologically grounded, combining both qualitative (focus groups, expert review) and quantitative (CFA, reliability testing) components, which strengthen the overall rigor. However some issues must be improved:

Item reduction rationale: While respecification improved model fit, the theoretical justification for item removal (beyond statistical criteria) could be elaborated to ensure the conceptual fidelity of the final instrument.

Reliability scope: The internal consistency measures are sound, but the study would benefit from reporting confidence intervals and potentially including test-retest reliability to assess temporal stability.

Generalizability: Since the sample is limited to Tacna, more discussion is needed on how the instrument may (or may not) generalize to other regions of Peru with distinct cultural or linguistic contexts.

Practical implications: The manuscript could be strengthened by offering clearer pathways for how this instrument will be used in public health decision-making or intervention planning in Peru

-How culturally and linguistically appropriate is the adapted Spanish version of the HLS-COVID-Q22 for the Viñani population?

-How representative and engaged were the focus group participants involved in the linguistic adaptation?

-To what extent does the post-pandemic timing of the data collection influence participants’ health literacy perceptions and responses?

-How will the results of this adapted instrument inform future public health interventions or policy decisions in the Tacna region? It should be discussed.

There is a probable problem. The direct linguistic translation alone was insufficient to preserve the original questionnaire's psychometric structure in this new cultural context, necessitating adaptation beyond language to ensure validity. This may reflect cultural differences in interpreting certain items or a need for further qualitative refinement. How this impact has been considered?

Author Response

Dear Reviewer 2,

Thank you for your comments and please find attached our replies.

Best wishes

Round 2

Reviewer 1 Report

Comments and Suggestions for Authors

I read the revised version of this manuscript with interest and sincerely appreciate the authors’ efforts to respond to the comments received. It is clear that the manuscript has been substantially improved, particularly in terms of the clarity of the introduction, the completion of the methodology section, and the strengthening of psychometric validation.

The authors have succeeded in clearly stating the study’s objective in the introduction and have presented the international context in a more logical manner. Additionally, the details regarding the pilot sample, focus group, and linguistic adaptation process have been included and now offer a complete picture of the qualitative phase. From a statistical standpoint, the use of CFA with the DWLS estimator, along with internal consistency indicators (ordinal alpha, omega, CR), and the inclusion of discriminant and concurrent validity tests, add methodological value to the study.

However, despite these clear improvements, several essential points remain either unaddressed or only partially addressed and require further attention:

  1. No test–retest procedure was conducted, meaning that the instrument was not tested for temporal stability. While I understand the time and project constraints, I would have expected either an alternative solution or a clear intention to conduct this step in future research.
  2. There is no mention of sampling bias, although all 490 participants come from a single region (Tacna). This omission affects the generalizability of the results and should at least be acknowledged as a limitation.
  3. There is no discussion of self-reporting bias, even though the instrument relies exclusively on individual perceptions. In a socio-educationally vulnerable context, this aspect deserves attention.
  4. The exploratory factor analysis (EFA) was only partially applied. While the authors justify their reliance on CFA based on conceptual equivalence, a complete exploratory analysis (with rotation, explained variance, etc.) would have significantly strengthened the credibility of the results in the context of cross-cultural adaptation.
  5. The discussion section remains one of the manuscript’s weakest parts. Although a comparative paragraph was added and some implications are mentioned, the overall tone is still descriptive, repeating content from the methods section, and lacking genuine reflection on limitations, international comparisons, or practical applications. For instance, no concrete scenarios are offered regarding how the questionnaire could be used in the Peruvian healthcare system, in health education, or in public policy.

This manuscript has potential and has made significant progress compared to the previous version. It is clear that the authors are well-versed in the subject and methodology, and the proposed instrument is valuable. However, in order to be suitable for publication in an international journal with a medical and methodological profile, further revisions are needed—particularly regarding limitations, external validation, and practical applicability.

Reviewer 2 Report

Comments and Suggestions for Authors

Dear Editorial Team

The revised version is satisfying.

Best Regards.

Author Response

Thank you for all your comments.